# Naturalize Revolution: Unprecedented AI-Driven Precision in Skin Cancer Classification Using Deep Learning

**Mohamad Abou Ali** [1,2,3] **, Fadi Dornaika** [1,4,*] **, Ignacio Arganda-Carreras** [1,4,5,6] **, Hussein Ali** [2]
**and Malak Karaouni** [3]

1 Department of Computer Science and Artificial Intelligence, University of the Basque Country (UPV/EHU), Manuel Lardizabal, 1, 20018 San Sebastian, Spain; mohamad.abouali01@liu.edu.lb (M.A.A.); ignacio.arganda@ehu.eus (I.A.-C.)

2 Department of Biomedical Engineering, Beirut International University (BIU), Salim Salam, Mazraa, Beirut 14404, Lebanon; 11830134@students.liu.edu.lb

3 Department of Biomedical Engineering, Lebanese International University (LIU), Salim Salam, Mazraa, Beirut 14404, Lebanon; 21830297@students.liu.edu.lb

4 Ikerbasque, Basque Foundation for Science, Plaza Euskadi, 5, 48009 Bilbao, Spain

5 Donostia International Physics Center (DIPC), Manuel Lardizabal, 4, 20018 San Sebastian, Spain

6 Biofisika Institute (CSIC, UPV/EHU), Barrio Sarriena s/n, 48940 Leioa, Spain

* Correspondence: fadi.dornaika@ehu.eus

**Abstract:** Background: In response to the escalating global concerns surrounding skin cancer, this study aims to address the imperative for precise and efficient diagnostic methodologies. Focusing on the intricate task of eight-class skin cancer classification, the research delves into the limitations of conventional diagnostic approaches, often hindered by subjectivity and resource constraints. The transformative potential of Artificial Intelligence (AI) in revolutionizing diagnostic paradigms is underscored, emphasizing significant improvements in accuracy and accessibility. Methods: Utilizing cutting-edge deep learning models on the ISIC2019 dataset, a comprehensive analysis is conducted, employing a diverse array of pre-trained ImageNet architectures and Vision Transformer models. To counteract the inherent class imbalance in skin cancer datasets, a pioneering "Naturalize" augmentation technique is introduced. This technique leads to the creation of two indispensable datasets—the Naturalized 2.4K ISIC2019 and groundbreaking Naturalized 7.2K ISIC2019 datasets—catalyzing advancements in classification accuracy. The "Naturalize" augmentation technique involves the segmentation of skin cancer images using the Segment Anything Model (SAM) and the systematic addition of segmented cancer images to a background image to generate new composite images. Results: The research showcases the pivotal role of AI in mitigating the risks of misdiagnosis and under-diagnosis in skin cancer. The proficiency of AI in analyzing vast datasets and discerning subtle patterns significantly augments the diagnostic prowess of dermatologists. Quantitative measures such as confusion matrices, classification reports, and visual analyses using Score-CAM across diverse dataset variations are meticulously evaluated. The culmination of these endeavors resulted in an unprecedented achievement—100% average accuracy, precision, recall, and F1-score—within the groundbreaking Naturalized 7.2K ISIC2019 dataset. Conclusion: This groundbreaking exploration highlights the transformative capabilities of AI-driven methodologies in reshaping the landscape of skin cancer diagnosis and patient care. The research represents a pivotal stride towards redefining dermatological diagnosis, showcasing the remarkable impact of AI-powered solutions in surmounting the challenges inherent in skin cancer diagnosis. The attainment of 100% across crucial metrics within the Naturalized 7.2K ISIC2019 dataset serves as a testament to the transformative capabilities of AI-driven approaches in reshaping the trajectory of skin cancer diagnosis and patient care. This pioneering work paves the way for a new era in dermatological diagnostics, heralding the dawn of unprecedented precision and efficacy in the identification and classification of skin cancers.

**Keywords:** convolutional neural net (CNN); vision transformer (ViT); ImageNet models; transfer learning (TL); machine learning (ML); deep learning (DP); skin cancer; naturalize; Segment Anything Model (SAM)

## 1. Introduction

Skin cancer, a widespread and potentially life-threatening disease, impacts millions globally. Its harmful effects can range from disfigurement to significant medical expenses, and even mortality if not diagnosed and treated early. Approximately one in five Americans are projected to develop skin cancer in their lifetime, with around 9500 daily diagnoses in the U.S. [1]. Beyond physical consequences, skin cancer can induce emotional distress due to invasive treatments and visible scars.

Skin cancer is a prevalent malignancy linked to prolonged exposure to ultraviolet (UV) radiation, either from the sun or artificial sources [2]. UV radiation causes DNA damage, leading to genetic mutations and abnormal cell growth. Fair-skinned individuals with a history of sunburns, especially in childhood, are more susceptible. Genetic factors, including familial cases and specific conditions like xeroderma pigmentosum, elevate risk. Aging, immune system suppression (in transplant recipients or HIV/AIDS patients), and certain chemical exposures also contribute. Individuals with prior skin cancer require vigilant follow-up and skin checks due to an increased risk of recurrence.

Figure 1 [3] shows different skin cancer stages from stage 0 to stage 4 and its corresponding severity.

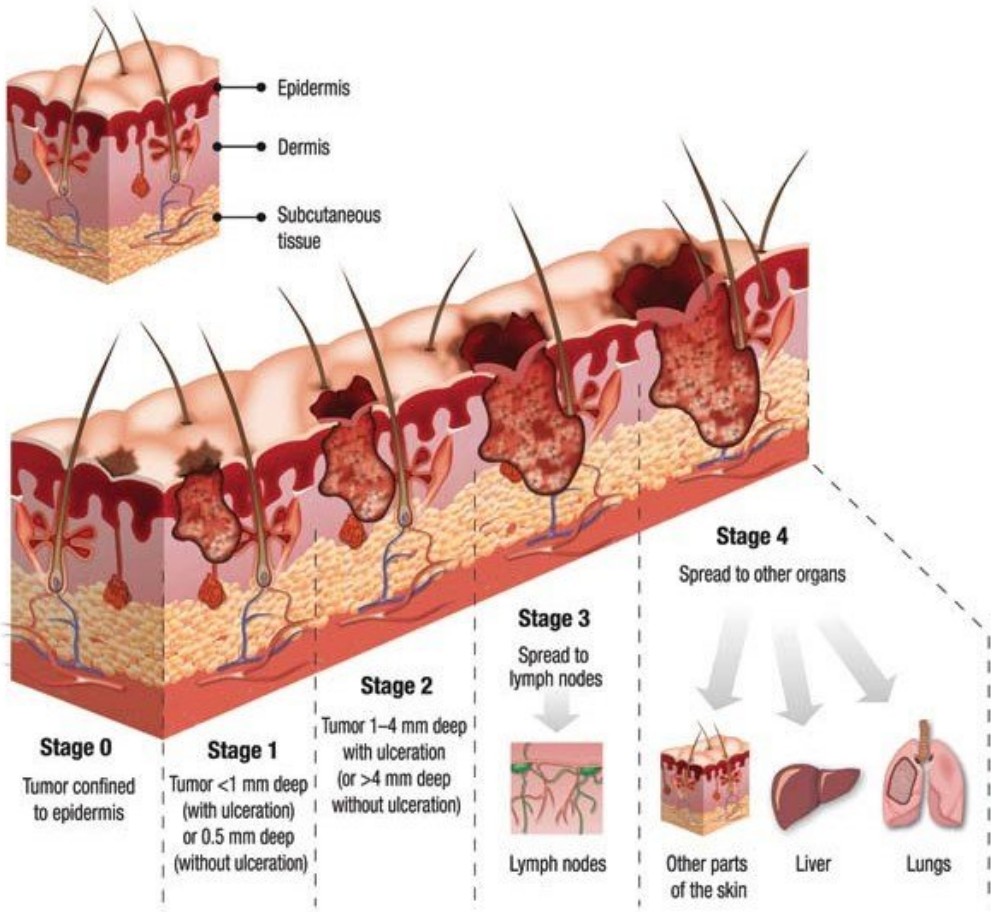

**Figure 1.** Skin cancer stages and severity.

The ISIC 2019 dataset [4] is a significant compilation within the International Skin Imaging Collaboration (ISIC) series, specifically curated for advancing research in dermatology, particularly in the field of computer-aided diagnosis (CAD) for skin cancer detection and classification. This dataset [4], released in 2019, is a continuation of the effort to provide a comprehensive collection of high-quality dermoscopic images accompanied by annotations and metadata. It consists of thousands of images showcasing various skin lesions, including melanomas, nevi, and other types of benign and malignant skin conditions.

One of the primary objectives of the ISIC 2019 dataset is to facilitate the development and evaluation of machine learning algorithms, computer vision models, and Artificial Intelligence systems geared towards accurate and early detection of skin cancers. Researchers, data scientists, and developers leverage this dataset to train, validate, and test their algorithms for automated skin lesion analysis, classification, and diagnosis. The availability of annotated images within the ISIC 2019 dataset [4] allows for supervised learning approaches, enabling algorithms to learn patterns and features associated with different types of skin lesions. By utilizing this dataset, researchers aim to improve the accuracy and efficiency of diagnostic tools, potentially aiding dermatologists and healthcare professionals in making more precise and timely diagnoses.

In recent years, deep learning [5] has brought about a transformative revolution in the field of machine learning. It stands out as the most advanced subfield, centering on artificial neural network algorithms inspired by the structure and function of the human brain. Deep learning techniques find extensive application in diverse domains, including but not limited to speech recognition, pattern recognition, and bioinformatics. Notably, in comparison to traditional machine learning methods, deep learning systems have demonstrated remarkable achievements in these domains. Recent years have witnessed the adoption of various deep learning strategies for computer-based medical applications [6], such as skin cancer detection. This paper delves comprehensively into the examination and evaluation of deep learning-based skin cancer classification techniques.

Our approach incorporates state-of-the-art deep learning models, including ImageNet ConvNets [7] and Vision Transformer (ViT) [8], through techniques like transfer learning, and fine-tuning. Evaluation encompasses quantitative assessments using confusion matrices, classification reports, and visual evaluations using tools like Score-CAM [9].

The integration of "Naturalize" techniques, as referenced in [10], alongside these strides represents significant headway in automating the analysis of skin cancer classification.

A consequence of employing the Naturalize technique is the establishment of two well-balanced datasets, namely Naturalized 2.4K and 7.2K datasets, encompassing 2400 and 7200 images, respectively, for each of the eight types of skin cancer. This paper extensively explores the methodologies and outcomes derived from these state-of-the-art approaches, shedding light on their transformative capacity within the realm of skin cancer.

After this introduction, the rest of the paper will continue as follows: Section 2 highlights the relevant literature related to the detection and classification of skin cancer using pre-trained CNNs, and Section 3 describes the methodology used in this study. In addition, Section 4 presents the experimental results obtained using pre-trained models and Google ViT for the skin cancer classification; an in-depth analysis of the results is performed. Finally, the paper is concluded in Section 5.

## 2. Related Works

Recent advancements in deep learning models for skin lesion classification have showcased significant progress. This review consolidates findings from notable studies employing diverse convolutional neural network (CNN) architectures for this purpose. These studies explore methodologies and performances using the ISIC2019 dataset.

Kassem et al. [11] utilized a GoogleNet (Inception V1) model with transfer learning on the ISIC2019 dataset, achieving 94.92% accuracy. They demonstrated commendable performance in recall (79.80%), precision (80.36%), and F1-score (80.07%).

Sun et al. [12] employed an Ensemble CNN-EfficientNet model on the ISIC2019 dataset, achieving an accuracy of 89.50%. Additionally, the authors investigated the integration of extra patient information to improve the precision of skin lesion classification. They presented performance metrics with recall (89.50%), precision (89.50%), and F1-score (89.50%).

Singh et al. [13] utilized the Ensemble Inception-ResNet model on the ISIC2019 dataset, achieving an accuracy of 96.72%. Their results showcased notable performance in recall (95.47%), precision (84.70%), and F1-score (89.76%).

In 2022, Li et al. [14] introduced the Quantum Inception-ResNet-V1, achieving 98.76% accuracy on the same ISIC2019 dataset. Their model exhibited substantial improvements in recall (98.26%), precision (98.40%), and F1-score (98.33%), signifying a significant leap in accuracy.

Mane et al. [15] leveraged MobileNet with transfer learning, achieving an accuracy of 83% on the ISIC2019 dataset. Despite relatively lower results compared to other models, their consistent performance across recall, precision, and F1-score at 83% highlighted robust classification.

Hoang et al. [16] introduced the Wide-ShuffleNet combined with segmentation techniques, achieving an accuracy of 84.80%. However, their model showed comparatively lower metrics for recall (70.71%), precision (75.15%), and F1-score (72.61%) than prior studies.

In 2023, Fofanah et al. [17] introduced a four-layer DCNN model, achieving an accuracy of 84.80% on a modified dataset split. Their model showcased well-rounded performance with a recall of 83.80%, precision of 80.50%, and an F1-score of 81.60%.

Similarly, Alsahaf et al. [18] proposed a Residual Deep CNN model in the same year, attaining an impressive accuracy of 94.65% on a different dataset split. They maintained equilibrium across metrics, with a recall of 70.78%, precision of 72.56%, and an F1-score of 71.33%.

Venugopal et al. [19] presented a modified version of the EfficientNetV2 model in 2023, achieving a high accuracy of 95.49% on a different dataset split. They demonstrated balance in key metrics, including recall (95%), precision (96%), and an F1-score of 95%.

Tahir et al. [20] proposed a DSCC-Net model with SMOTE Tomek in 2023, achieving an accuracy of 94.17% on a different dataset split. Their model exhibited well-balanced metrics, with a recall of 94.28%, precision of 93.76%, and an F1-score of 93.93%.

Radhika et al. [21] introduced an MSCDNet Model in 2023, achieving an outstanding accuracy of 98.77% on a different dataset split. Their model maintained a harmonious blend of metrics, with a recall of 98.42%, precision of 98.56%, and an F1-score of 98.76%.

These studies collectively showcase the evolution of skin lesion classification models, indicating significant progress in accuracy and performance metrics. Comparative analysis highlights the strengths and weaknesses of each model, laying the groundwork for further advancements in dermatological image classification.

The literature review focuses on a series of studies (Table 1), concentrating on automating skin cancer classification using the ISIC2019 dataset, offering a summarized view of these endeavors.

Our groundbreaking research presents the novel augmentation technique "Naturalize", specifically designed to tackle the challenges posed by data scarcity and class imbalance within deep learning. Through the implementation of "Naturalize", we have successfully overcome these hurdles, achieving an unprecedented 100% average testing accuracy, precision, recall, and F1-score in our skin cancer classification model. This groundbreaking technique revolutionizes the landscape of deep learning, offering a solution that not only elevates classification performance but also redefines the potential for accurate and reliable diagnosis across various imbalanced skin cancer classes.

**Table 1.** Overview of related work.

| Ref. | Model and Approach | Dataset | Split Ratio | Accuracy | Recall | Precision | F1-Score |
|------|--------------------|---------|-------------|----------|--------|-----------|----------|
| [11] | GoogleNet (Inception V1) and Transfer Learning | ISIC2019 | 80/10/10 | 94.92 | 79.80 | 80.36 | 80.07 |
| [12] | Ensemble CNN-EfficientNet | ISIC2019 | 75/25 | 89.5 | 89.5 | 89.5 | 89.5 |
| [13] | Ensemble Inception-ResNet | ISIC2019 | 60/20/20 | 96.72 | 95.47 | 84.70 | 89.76 |
| [14] | Quantum Inception-ResNet-V1 | ISIC2019 | 80/10/10 | 98.76 | 98.26 | 98.40 | 98.33 |
| [15] | MobileNet and Transfer Learning | ISIC2019 | 80/10/10 | 83 | 83 | 83 | 82 |
| [16] | Wide ShuffleNet and Segmentation | ISIC2019 | 90/10 | 84.80 | 70.71 | 75.15 | 72.61 |
| [17] | Four-layer DCNN | ISIC2019 | 60/10/30 | 84.80 | 83.80 | 80.50 | 81.60 |
| [18] | Residual Deep CNN Model | ISIC2019 | 70/15/15 | 94.65 | 70.78 | 72.56 | 71.33 |
| [19] | Modified EfficientNetV2 | ISIC2019 | 80/20 | 95.49 | 95 | 96 | 95 |
| [20] | DSCC-Net with SMOTE Tomek | ISIC2019 | 80/10/10 | 94.17 | 94.28 | 93.76 | 93.93 |
| [21] | MSCDNet Model | ISIC2019 | 70/20/10 | 98.77 | 98.42 | 98.56 | 98.76 |

## 3. Materials and Methods

In this section, we offer an in-depth explanation of our methodology for classifying skin cancer images using the challenging ISIC2019 dataset. The steps of our approach are visually depicted in Figure 2.

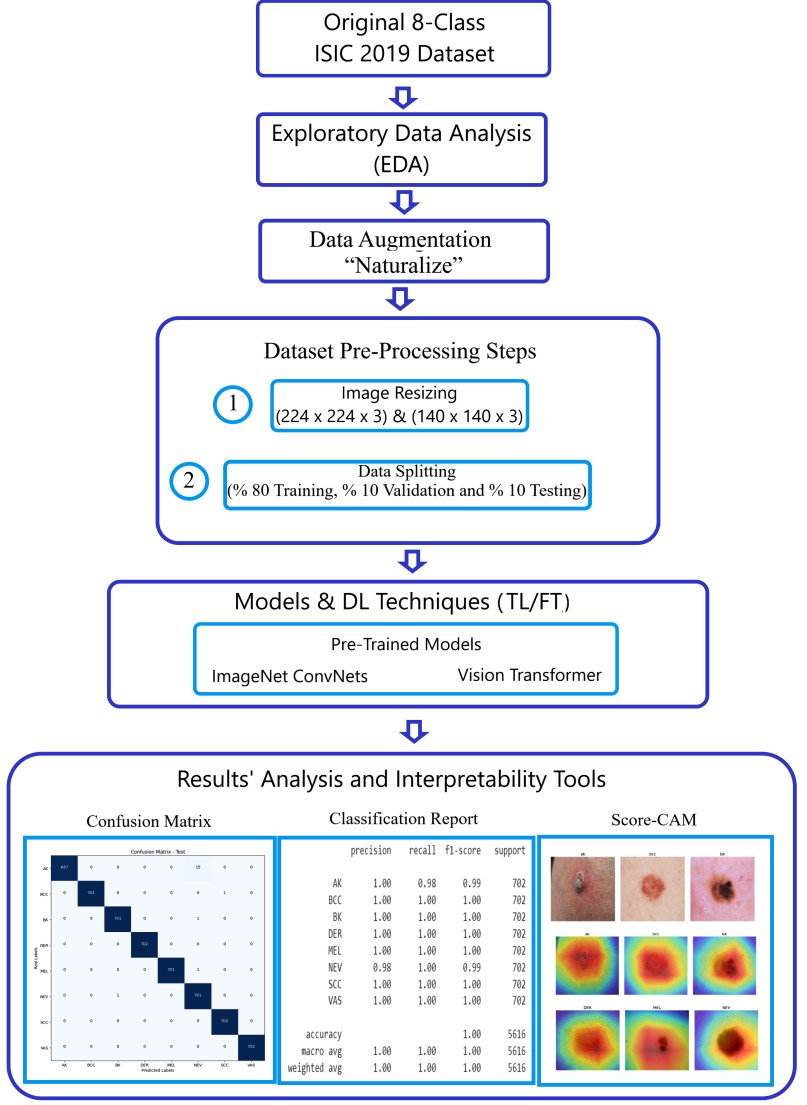

**Figure 2.** Methodology workflow using the ISIC 2019 dataset.

### 3.1. ISIC-2019 Dataset

3.1.1. Original 8-Class ISIC 2019 Dataset

The initial ISIC 2019 dataset [4], obtained from an online repository, consists of 25,331 images categorized into eight distinct classes representing different types of skin cancer. These classes are Actinic Keratosis (AK), Basal Cell Carcinoma (BCC), Benign Keratosis (BK), Dermatofibroma (DER), Melanocytic Nevi (NEV), Melanoma (MEL), Vascular Skin Lesion (VAS), and Squamous Cell Carcinoma (SCC).

To address the unbalanced distribution of images within the original ISIC-2019 dataset, we modified it by reducing the number of images for three types of skin cancer (MEL, NV, BCC) to 2.4k, aligning them with the existing count of 2.4k images for the BK type. This adjustment was made to achieve balance among the different cancer types. We applied the Naturalize Augmentation technique during this process. Consequently, the updated dataset now comprises 19,200 balanced images across the eight types of skin cancer.

Table 2 [4] provides an overview of the distribution of the eight skin cancer classes within the original ISIC 2019 dataset.

**Table 2.** Summary of the ISIC-2019 dataset.

| Number | Cell Type | Total of Images by Type | Percent |
|---|---|---|---|
| 1 | Actinic Keratosis | 867 | 3.322 |
| 2 | Basal Cell Carcinoma | 3323 | 13.11 |
| 3 | Benign Keratosis | 2624 | 10.35 |
| 4 | Dermatofibroma | 239 | 0.94 |
| 5 | Melanocytic Nevi | 12,875 | 50.82 |
| 6 | Melanoma | 4522 | 17.85 |
| 7 | Vascular Skin Lesion | 253 | 1.138 |
| 8 | Squamous Cell Carcinoma | 628 | 2.47 |
| | Total | 25,331 | 100 |

The images in the ISIC dataset adhere to a standard size of 1024 × 1024 pixels [4], which needs to be resized into "224 × 224" and "140 × 140" to make the use of it more flexible in the work.

Figure 3 shows the 8 types of skin cancer found in the original ISIC2019 dataset.

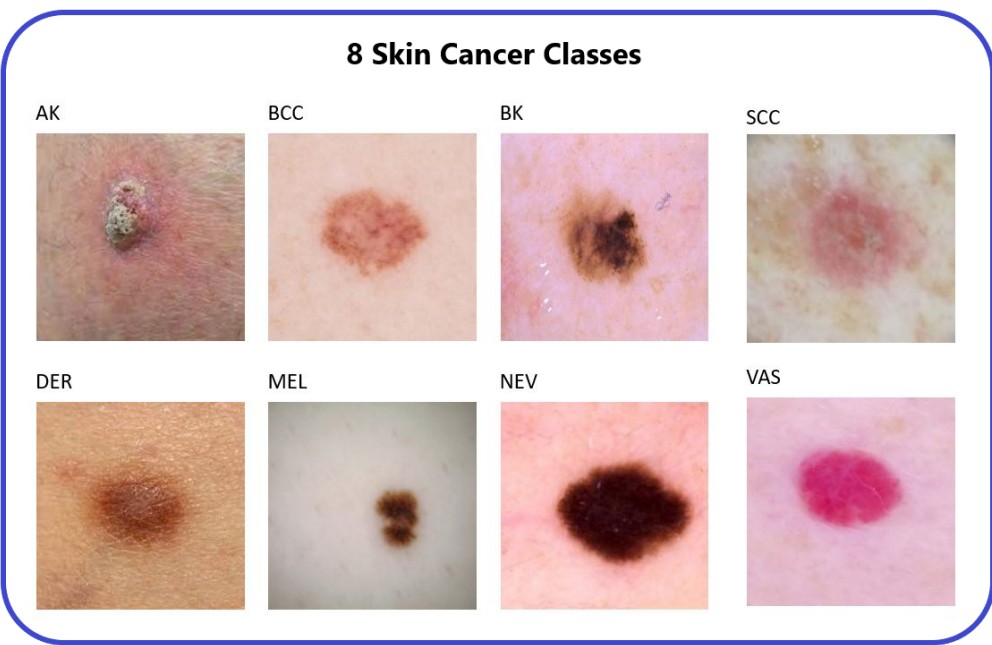

**Figure 3.** The 8 types of skin cancer [14].

### 3.1.2. Pruned 2.4K ISIC2019

Due to substantial variations in the quantity of available images, it was necessary to reduce the number of photos in specific categories. This adjustment aimed to alleviate the pronounced differences among various types of skin cancer.

Table 3 summarizes the distribution of the Pruned 2.4K ISIC2019 dataset in the 8 classes.

**Table 3.** Summary of the Pruned 2.4K ISIC-2019 dataset.

| Number | Cell Class | Symbol | Images by Class | (%) |
|:---:|:---|:---|---:|---:|
| 1 | Actinic Keratosis | AK | 867 | 7.5 |
| 2 | Basal Cell Carcinoma | BCC | 2400 | 20.7 |
| 3 | Benign Keratosis | BK | 2400 | 20.7 |
| 4 | Dermatofibroma | DER | 239 | 2 |
| 5 | Melanocytic Nevi | NEV | 2400 | 20.7 |
| 6 | Melanoma | MEL | 2400 | 20.7 |
| 7 | Vascular Skin Lesion | VAS | 253 | 2.1 |
| 8 | Squamous Cell Carcinoma | SCC | 628 | 5.6 |
| | Total | | 11,587 | 100 |

### 3.1.3. Naturalized 2.4K and 7.2K ISIC2019 Datasets

Our goal was to achieve an equal number of photos across all eight types of skin cancer. The Naturalize augmentation is employed to achieve this target. Two balanced updated version of ISIC2019 are created using the Naturalize augmentation technique: Naturalized 2.4K ISIC2019 and Naturalized 7.2K ISIC2019 datasets.

Table 4 summarizes the distribution of the Naturalized 2.4K ISIC2019 dataset in the 8 classes.

**Table 4.** Summary of the Naturalized 2.4K ISIC-2019 dataset.

| Number | Cell Class | Symbol | Images by Class | (%) |
|:---:|:---|:---|---:|---:|
| 1 | Actinic Keratosis | AK | 2400 | 12.5 |
| 2 | Basal Cell Carcinoma | BCC | 2400 | 12.5 |
| 3 | Benign Keratosis | BK | 2400 | 12.5 |
| 4 | Dermatofibroma | DER | 2400 | 12.5 |
| 5 | Melanocytic Nevi | NEV | 2400 | 12.5 |
| 6 | Melanoma | MEL | 2400 | 12.5 |
| 7 | Vascular Skin Lesion | VAS | 2400 | 12.5 |
| 8 | Squamous Cell Carcinoma | SCC | 2400 | 12.5 |
| | Total | | 19,200 | 100 |

Table 5 summarizes the distribution of the Naturalized 7.2K ISIC2019 dataset in the 8 classes.

**Table 5.** Summary of the Naturalized 7.2K ISIC-2019 dataset.

| Number | Cell Class | Symbol | Images by Class | (%) |
|:---:|:---|:---|---:|---:|
| 1 | Actinic Keratosis | AK | 7200 | 12.5 |
| 2 | Basal Cell Carcinoma | BCC | 7200 | 12.5 |
| 3 | Benign Keratosis | BK | 7200 | 12.5 |
| 4 | Dermatofibroma | DER | 7200 | 12.5 |
| 5 | Melanocytic Nevi | NEV | 7200 | 12.5 |
| 6 | Melanoma | MEL | 7200 | 12.5 |
| 7 | Vascular Skin Lesion | VAS | 7200 | 12.5 |
| 8 | Squamous Cell Carcinoma | SCC | 7200 | 12.5 |
| | Total | | 57,600 | 100 |

The Naturalize augmentation technique can generate any number of skin cancer images with unique content and quality resembling the original ISIC2019 dataset. This is achieved through the benefit from the power of randomness of the addition of segmented skin cancer images with different skin backgrounds.

### 3.2. Exploratory Data Analysis (EDA)

Exploratory Data Analysis (EDA) was conducted to gain insights into the nature of the dataset. This involved training and testing pre-trained ImageNet models with the original ISIC 2019 dataset, analyzing the confusion matrix, and generating a classification report.

The primary observation from the exploratory data analysis (EDA) reveals a significant influence stemming from extreme class imbalances, notably in categories such as DER and NEV, within the ISIC 2019 dataset. This imbalance markedly impacts the overall performance metrics (average accuracy, precision, recall, and F1-score).

Addressing this issue is where the "Naturalize" augmentation technique comes into play. This technique involves generating new images for classes that have insufficient representation, maintaining a quality that mirrors the original images. As a result, "Naturalize" effectively resolves the pronounced imbalances among classes while preserving image quality.

### 3.3. Data Augmentation "Naturalize"

The pseudocode shown in Algorithm 1 demonstrates the principle behind the "Naturalize" augmentation technique and how it works.

---

**Algorithm 1** Naturalize algorithm.

---

 1: **Imports and Paths**
 2: Import os, random, image_processing, SAM_model
 3: Define file paths and import essential libraries
 4: **Load SAM_model and ISIC 2019 Dataset**
 5: Mount Google_drive
 6: Load Skin Cancer images from ISIC 2019 dataset
 7: SAM = load_model(SAM_model)
 8: **Segment ISIC 2019 Dataset Using SAM**
 9: Segment ISIC 2019 images using SAM into segmented "Cancer" images
10: Save segmented "Cancer" images into "Skin Cancer" dataset on Google_drive
11: **Random Selection from Skin Background dataset**
12: Select randomly Skin Background image from Skin Background dataset
13: **Composite Image Creation**
14: **for** i in range(num_images) **do**
15:     Load Skin Background image
16:     Select randomly "Cancer" image from "Skin Cancer" dataset
17:     Rotate randomly "Cancer" image
18:     Add "Cancer" image at random position to Skin Background image
19:     Save the composite image on Google_drive
20: **end for**

---

The "Naturalize" augmentation technique consists of two primary steps:

- Step 1—Segmentation Figure 4:
  Within the ISIC 2019 dataset, images depicting four different types of skin cancer were divided into smaller sets through the application of the "Segment Anything Model (SAM)" developed by Meta AI [22]. This process produced segments for AK, DER, VAS, and SCC. The inclusion of these new images in the classes positively influenced the accuracy of classification as evidenced by the performance metrics and classification report from the prior EDA analysis.

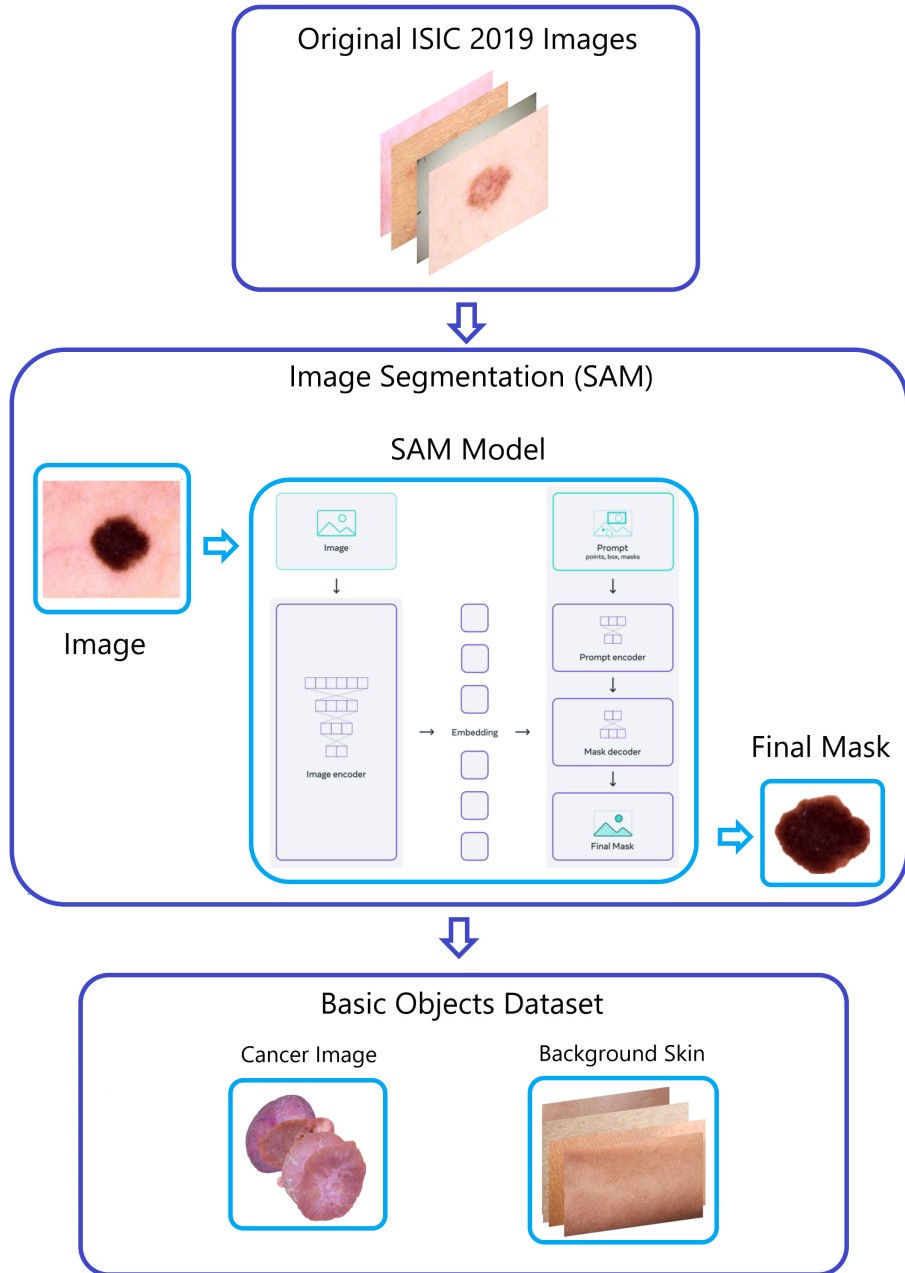

**Figure 4.** The "Naturalize" first step—segmentation.

- Step 2—Generating Composite Images (Figure 5):
  To produce composite images, we merged the four SAM-segmented categories with randomly chosen photographs of healthy skin within the respective sub-datasets (AK, DER, VAS, and SCC). This procedure is visually demonstrated in Figures 4 and 5, using the creation of composite skin cancer images as an example.

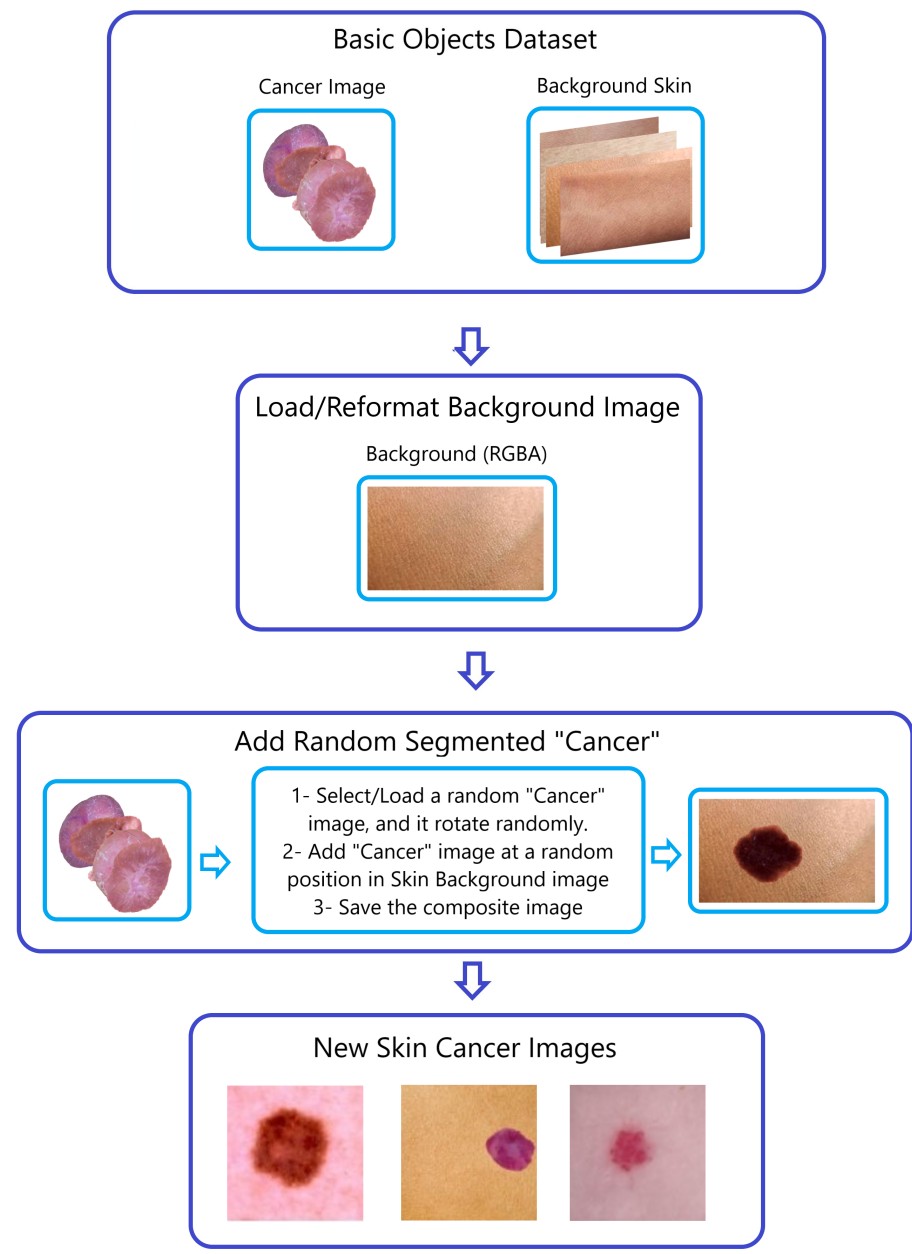

**Figure 5.** The "Naturalize" second step—composite image generation.

Guided by our meticulous exploratory data analysis (EDA), we have judiciously pruned select images from multiple classes, embodying our unwavering commitment to data quality. Following the incorporation of images, the quantity of skin cancer images within the initial 8-class ISIC2019 dataset saw a significant rise. Through the initial utilization of the "Naturalize" technique, the number of skin cancer images surged from 1987 images to a substantial 9600 images. TABLE IV effectively portrays the remarkable evolution of the original 1987 skin cancer images referring to (AK, DER, VAS, SCC) in ISIC2019 dataset into 9600 skin cancer images spanning four different types (AK, DER, VAS, and SCC).

The dataset experienced significant growth due to the implementation of the "Naturalize" augmentation method, resulting in the development of the ISIC2019 dataset with approximately 9.6K images. This expansion was achieved by adding between 1500 to 2000 images to each of the four sub-datasets representing the following skin cancer classes: AK, DER, VAS, and SCC.

The choice to exclusively incorporate the skin cancer categories "AK, DER, VAS, and SCC" in the "Naturalize" applications stems from the findings in the classification report. This decision is driven by the goal of enhancing the overall precision, recall, F1-score, and accuracy averages.

### 3.4. Comparison between Naturalize and Conventional Augmentation Techniques

Conventional image augmentation [23,24] commonly involves basic transformations like rotation, flipping, and color adjustments to enhance datasets by introducing variety through general image manipulations. In contrast, the "Naturalize" augmentation method is characterized by its complexity and specificity. This method employs a targeted segmentation process using the "Segment Anything Model" to isolate specific object classes within the dataset. For example, in the original ISIC2019 dataset, "Naturalize" isolates images of foreground skin cancers, such as Ak, BCC, BK, DER, NEV, MEL, VAS, and SCC, from the background skin images.

The application of the SAM model to the original ISIC2019 dataset yields a significantly large number of segmented foreground skin cancer instances. The random incorporation of these segmented objects results in an extensive array of unique and realistic replicas of the original ISIC2019 dataset. Importantly, this enables the addition of different segmented skin cancer images to various background skin images with diverse skin colors, generating new, previously non-existent skin cancer images while preserving the original ISIC2019 image quality.

Furthermore, the versatility of the "Naturalize" technique extends beyond medical imaging. Through the segmentation and reintroduction of all objects in the original images into background images, "Naturalize" can be applied to various applications, both within and beyond the medical field. This adaptability underscores its potential for widespread use, showcasing its applicability beyond medical image augmentation.

Crucially, "Naturalize" maintains the realism of skin cancer sizes, preserving the authentic dimensions of the original ISIC2019 images. In summary, the focus of "Naturalize" is on both authenticity and diversity in medical images, tailoring the augmentation process to specific requirements rather than relying on generic transformations.

### 3.5. Naturalized 2.4K ISIC2019 and Naturalized 7.2K ISIC2019 Datasets Preprocessing

The preprocessing of the Naturalized 2.4K ISIC2019 and Naturalized 7.2K ISIC2019 datasets involved two primary steps:

- Step 1—Image Resizing: The images were resized to match the standard "224 × 224" image input size required by pre-trained ImageNet ConvNets and ViT models. Additionally, the images were resized to dimensions of "140 × 140", aiming to optimize computational resources, especially with a sizable dataset like the Naturalized 7.2K ISIC2019 dataset.
- Step 2—Data Splitting: The Naturalized 2.4K ISIC2019 and Naturalized 7.2K ISIC2019 datasets were split into three subsets: an 80% training set, a 10% validation set, and a 10% testing set.

### 3.6. Models and DL Techniques (TL/FT)

Two types of model architectures were utilized in this study: pre-trained ImageNet ConvNets, and pre-trained Vision Transformers (ViT). Additionally, two DL techniques [25] were employed to train the pre-trained models: transfer learning (TL) and fine-tuning (FT).

#### 3.6.1. Pre-Trained ImageNet ConvNets

Pre-trained ImageNet models are an explicit example of ConvNets, which are trained on a large dataset.

For this study, pre-trained ImageNet models formed the core of the research. Notable models utilized in this investigation included ConvNexTBase and ConvNeXtLarge [26], DenseNet-201 [27], EfficientNetV2 B0 [28], InceptionResNet [29], Xception [30], and VGG16 [31], and VGG-

19 [31]. Figure 6 offers an illustration of the VGG-19 [31] model's architecture when applied to skin cancer classification.

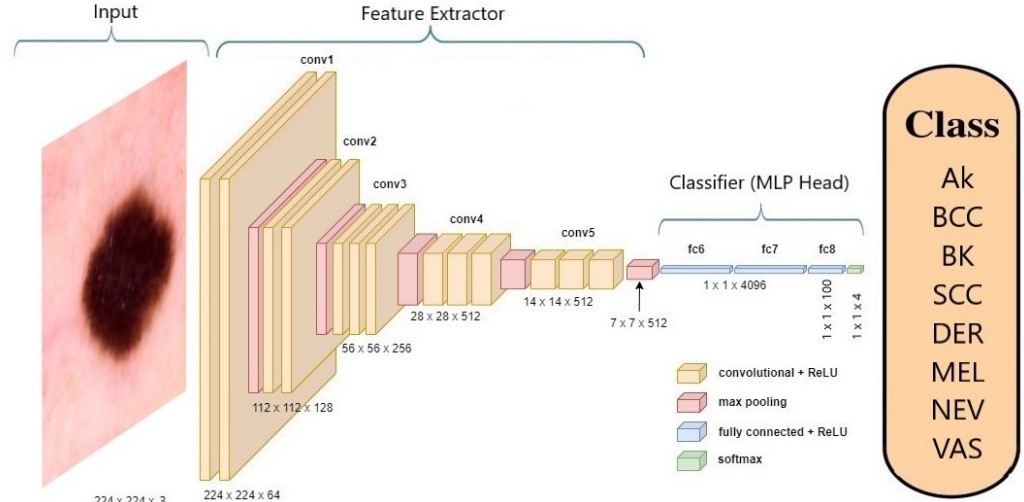

**Figure 6.** Architecture of VGG-19 model classifying a skin cancer image.

### 3.6.2. Pre-Trained Vision Transformer (ViT)

The study employed the Vision Transformer (ViT) [8] architecture, which is derived from the transformer architecture frequently utilized in Natural Language Processing (NLP). This approach entailed dividing input images into smaller patches and subjecting each patch to processing through a transformer encoder. In contrast to conventional convolutional layers, ViT employed self-attention mechanisms to extract features from the input images, enabling the model to analyze the entire image simultaneously. The research utilized the "ViT" configuration with 12 encoder blocks, and Figure 7 demonstrates its use in classifying skin cancer [8].

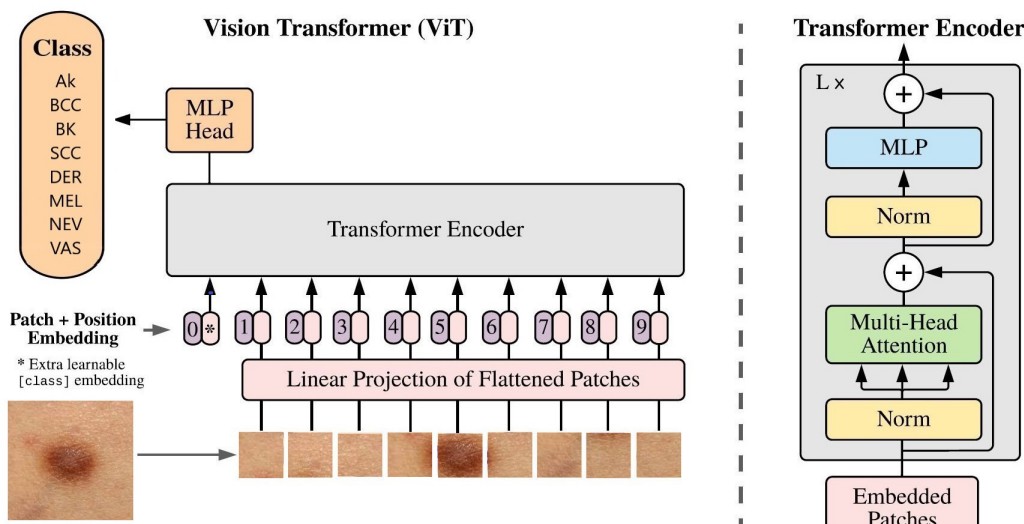

**Figure 7.** Architecture of ViT classifying a skin cancer image.

### 3.6.3. DL Techniques (TL/FT)

A pre-trained ImageNet model comprises a Convolutional Base, responsible for extracting features, and a classifier, which is a Multi-Layer Perceptron (MLP) head. In the context of transfer learning (TL) [32], the process involves replacing the MLP head with a

new one and then retraining the model on a specific dataset. During this transfer learning phase, the Convolutional Base remains fixed and not trainable.

When fine-tuning (FT) [33] is applied, both the Convolutional Base and the MLP head undergo further training, adjusting their parameters to suit a new learning task.

To achieve an optimal deep learning skin cancer tool, this work employs two deep learning techniques, namely, transfer learning (TL) and fine-tuning (FT).

### 3.7. Results' Analysis and Interpretability Tools

Apart from the accuracy metrics, which include accuracy and loss, three tools for analyzing and interpreting results are employed. These tools consist of the confusion matrix, classification reports, and Score-CAM.

### 3.7.1. Confusion Matrix

A confusion matrix [34], also known as an error matrix, provides a visual representation of how well an algorithm performs, particularly in supervised learning scenarios. It presents actual classes in the rows and predicted classes in the columns. Figure 8 illustrates such a matrix in a multi-class classification context, highlighting "TN and TP" for correctly identified negative and positive cases, and "FN and FP" for cases that were incorrectly classified.

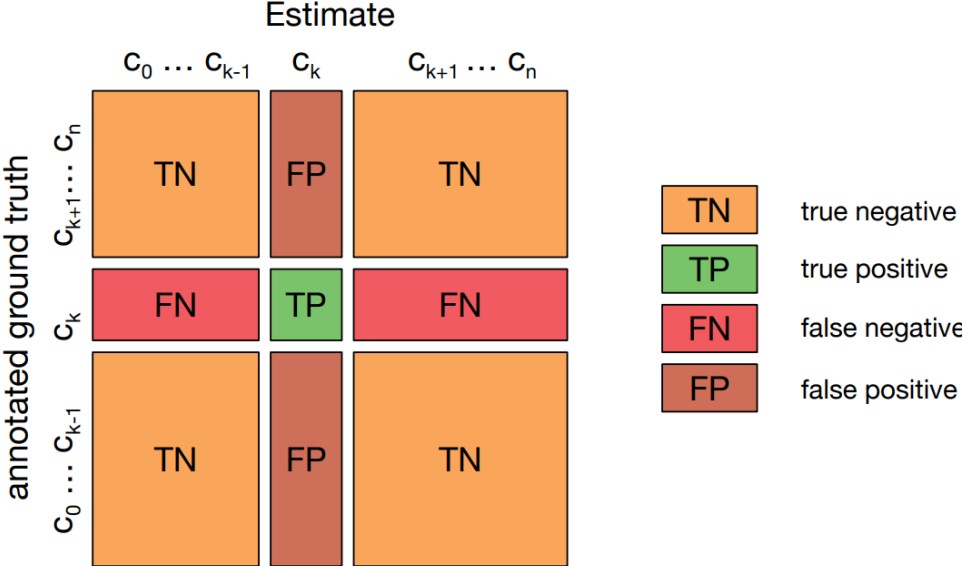

**Figure 8.** Confusion matrix for multiclass classification.

An illustrative numerical example of the confusion matrix is presented in Figure 9. This figure showcases the confusion matrix resulting from the fine-tuning of DenseNet201 with the Naturalized 7.2K 8-class ISIC2019 dataset.

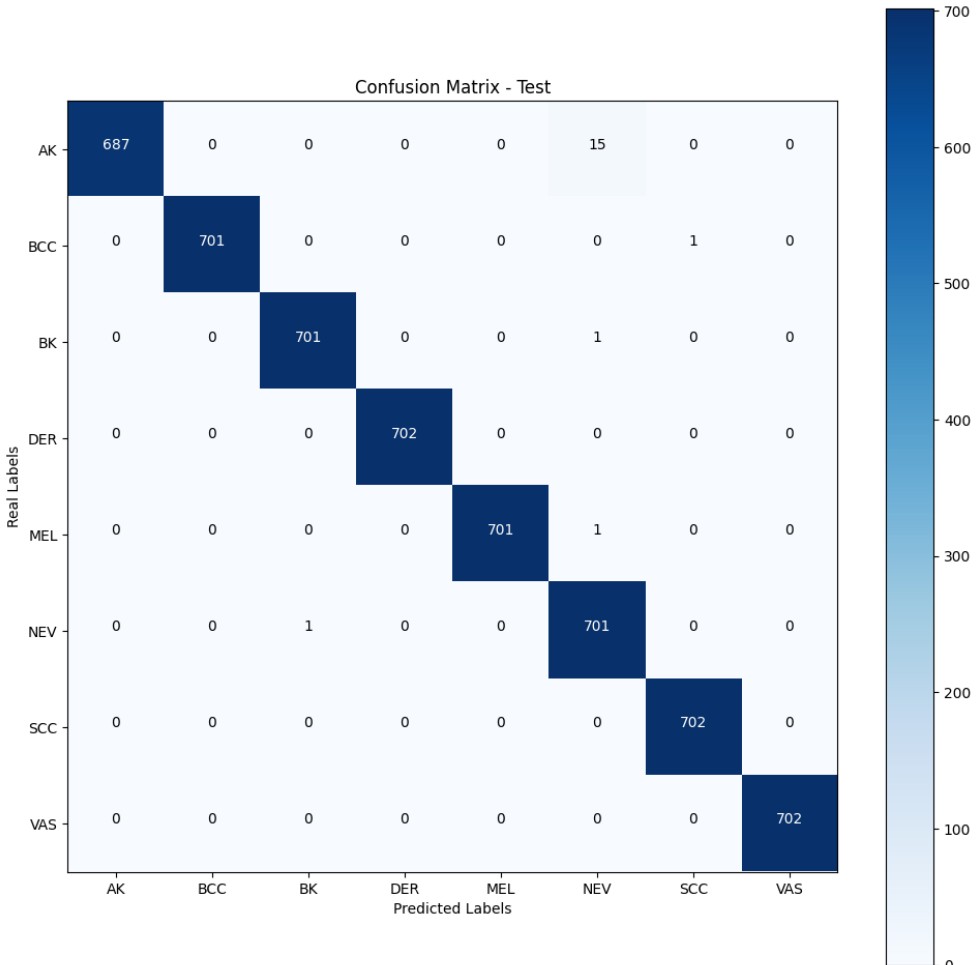

**Figure 9.** Confusion matrix—fine-tuned DenseNet201 with the Naturalized 7.2K dataset.

### 3.7.2. Classification Report

In the assessment, the evaluation of prediction quality relies on metrics such as precision, recall, and F1-score for individual classes. Additionally, it includes macro and weighted average accuracies to gauge overall performance. *Accuracy*, computed as a percentage of correct predictions, is determined by Equation (1) [34]:

$$Accuracy = \frac{TP + TN}{TP + TN + FP + FN} \tag{1}$$

*Precision* measures the quality of a positive prediction made by the model, and Equation (2) [34] demonstrates its computational process:

$$Precision = \frac{TP}{TP + FP} \tag{2}$$

*Recall* measures how many of the true positives (TPs) were recalled (found) and calculated using Equation (3) [34]:

$$Recall = \frac{TP}{TP + FN} \tag{3}$$

F1-Score is the harmonic mean of precision and recall and can be calculated using Equation (4) [34]:

$$F1 = \frac{2 * Precision * Recall}{Precision + Recall} = \frac{2 * TP}{2 * TP + FP + FN} \tag{4}$$

### 3.7.3. Score-CAM

A Score-CAM, as described in reference [9], is a visual explanation technique that assigns weights to scores using class activation mapping (CAM) in CNN models. It serves the purpose of providing insights into the inner workings of CNN models.

## 4. Results

This section offers a comprehensive summary of experiments focused on the 8-class classification of skin cancer. Various models, such as pre-trained ImageNet (ConvNextBase, ConvNeXtLarge, DenseNets, InceptionResNet V2, EfficientNetB0, VGG-19, VGG16, and Xception) and ViT models, were employed. The experiments were carried out using the challenging ISIC2019 dataset. To address class imbalance, the "Naturalize" augmentation technique was introduced, leading to the creation of two new balanced datasets named Naturalized 2.4K ISIC2019 and Naturalized 7.2K ISIC2019. The performance of the models was assessed quantitatively through confusion matrices and classification reports, and visually using Score-CAM on four types of datasets: original ISIC2019, updated ISIC2019, and Naturalized 2.4K ISIC2019 and Naturalized 7.2K ISIC2019 datasets.

### 4.1. Naturalized 2.4K ISIC2019 Dataset Results

Initially, all pre-trained models were fitted using transfer learning, but it was observed that fine-tuning led to better results. Table 6 presents the accuracy scores of the training, validation, and testing subsets of the Naturalized 2.4K ISIC2019 dataset for the fine-tuned models. Notably, the DenseNet201 model achieved the highest validation and accuracies, while the ConvNexTBase model recorded the highest training accuracy.

**Table 6.** Naturalized 2.4K ISIC 2019—summary of models' training, validation, and testing accuracies.

| Model | Accuracy | | |
| --- | --- | --- | --- |
| | Training | Validation | Testing |
| ConvNexTBase | 0.99 | 0.95 | 0.92 |
| ConvNeXtLarge | 0.87 | 0.84 | 0.84 |
| DenseNet-201 | 0.97 | 0.95 | 0.95 |
| EfficientNetV2 B0 | 0.88 | 0.85 | 0.82 |
| InceptionResNetV2 | 0.94 | 0.90 | 0.89 |
| VGG16 | 0.97 | 0.93 | 0.94 |
| VGG-19 | 0.96 | 0.89 | 0.90 |
| ViT | 0.89 | 0.87 | 0.90 |
| Xception | 0.94 | 0.91 | 0.82 |

Table 7 provides the macro-average precision, recall, and F1-score of the testing subset of the Naturalized 2.4K ISIC2019 dataset for the fine-tuned models, with the DenseNet-201 model achieving the best results.

**Table 7.** Naturalized 2.4K ISIC 2019—summary of models' macro-average precision, recall, and F1-scores.

| Model | Macro Average | | |
|---|---|---|---|
| | **Precision** | **Recall** | **F1-Score** |
| ConvNexTBase | 0.93 | 0.92 | 0.91 |
| ConvNeXtLarge | 0.87 | 0.86 | 0.87 |
| DenseNet-201 | 0.96 | 0.95 | 0.95 |
| EfficientNetV2 B0 | 0.86 | 0.82 | 0.80 |
| InceptionResNetV2 | 0.90 | 0.89 | 0.88 |
| VGG16 | 0.94 | 0.94 | 0.94 |
| VGG-19 | 0.90 | 0.90 | 0.89 |
| ViT | 0.91 | 0.90 | 0.90 |
| Xception | 0.86 | 0.87 | 0.86 |

Given the superior performance of the DenseNet-201model in the validation and testing subsets, it was selected for subsequent trials.

*4.2. DenseNet-201*

Table 8 presents the classification report of the fine-tuned DenseNet-201 model using the original ISIC 2019 dataset.

**Table 8.** DenseNet201—classification report for the original ISIC 2019.

| Class | Precision | Recall | F1-Score | Support |
|---|---|---|---|---|
| AK | 0.61 | 0.67 | 0.60 | 66 |
| BCC | 0.74 | 0.69 | 0.79 | 333 |
| BK | 0.58 | 0.88 | 0.79 | 263 |
| DER | 0.56 | 0.75 | 0.69 | 24 |
| NEV | 0.88 | 0.93 | 0.90 | 1287 |
| MEL | 0.65 | 0.36 | 0.46 | 452 |
| VAS | 0.85 | 0.87 | 0.89 | 63 |
| SCC | 0.75 | 0.94 | 0.86 | 25 |
| Accuracy | | | 0.78 | 2513 |
| Macro Avg. | 0.76 | 0.68 | 0.70 | 2513 |
| Weighted Avg. | 0.85 | 0.81 | 0.81 | 2513 |

Table 9 displays the classification report of the fine-tuned DenseNet201 model using the Pruned 2.4K ISIC 2019 dataset.

**Table 9.** DenseNet201—classification report for the Pruned 2.4K ISIC 2019.

| Class | Precision | Recall | F1-Score | Support |
|---|---|---|---|---|
| AK | 0.55 | 0.67 | 0.60 | 66 |
| BCC | 0.92 | 0.69 | 0.79 | 240 |
| BK | 0.71 | 0.88 | 0.79 | 240 |
| DER | 0.64 | 0.75 | 0.69 | 24 |
| NEV | 0.75 | 0.25 | 0.38 | 240 |
| MEL | 0.57 | 0.82 | 0.67 | 240 |
| VAS | 0.57 | 0.87 | 0.69 | 63 |
| SCC | 0.77 | 0.96 | 0.86 | 25 |
| Accuracy | | | 0.68 | 1138 |
| Macro Avg. | 0.69 | 0.74 | 0.68 | 1138 |
| Weighted Avg. | 0.72 | 0.68 | 0.66 | 1138 |

Table 10 displays the classification report of the fine-tuned DenseNet201 model using the Naturalized 2.4K ISIC 2019 dataset.

**Table 10.** DenseNet201—classification report for the Naturalized 2.4K ISIC 2019 with the testing dataset sourced from the Naturalized 2.4K ISIC 2019.

| Class | Precision | Recall | F1-Score | Support |
|---|---|---|---|---|
| AK | 0.98 | 0.99 | 0.98 | 240 |
| BCC | 0.99 | 0.95 | 0.97 | 240 |
| BK | 0.93 | 0.97 | 0.95 | 240 |
| DER | 0.98 | 1.00 | 0.99 | 240 |
| NEV | 0.98 | 0.75 | 0.85 | 240 |
| MEL | 0.81 | 0.99 | 0.89 | 240 |
| VAS | 0.99 | 0.97 | 0.98 | 240 |
| SCC | 1.00 | 1.00 | 1.00 | 240 |
| Accuracy | | | 0.95 | 1920 |
| Macro Avg. | 0.96 | 0.95 | 0.95 | 1920 |
| Weighted Avg. | 0.96 | 0.95 | 0.95 | 1920 |

Table 11 showcases the classification report of the fine-tuned DenseNet201 model utilizing the Naturalized 2.4K ISIC 2019 dataset. The disparity observed between Tables 10 and 11 stems from the distinct origins of the testing dataset images. Specifically, Table 11 employs images solely sourced from the Original ISIC 2019 dataset for testing, whereas Table 10 exclusively uses images from the Naturalized 2.4K ISIC 2019 dataset.

**Table 11.** DenseNet201—classification report for the Naturalized 2.4K ISIC 2019 with the testing dataset sourced from the Original ISIC 2019.

| Class | Precision | Recall | F1-Score | Support |
|---|---|---|---|---|
| AK | 0.98 | 0.98 | 0.98 | 240 |
| BCC | 0.99 | 0.98 | 0.99 | 240 |
| BK | 0.95 | 1.00 | 0.97 | 240 |
| DER | 1.00 | 1.00 | 1.00 | 240 |
| NEV | 0.85 | 0.98 | 0.91 | 240 |
| MEL | 0.99 | 0.80 | 0.89 | 240 |
| VAS | 1.00 | 1.00 | 1.00 | 240 |
| SCC | 1.00 | 0.99 | 0.99 | 240 |
| Accuracy | | | 0.97 | 1920 |
| Macro Avg. | 0.97 | 0.97 | 0.97 | 1920 |
| Weighted Avg. | 0.97 | 0.97 | 0.97 | 1920 |

The classification reports of the fine-tuned DenseNet201 model in Tables 10 and 11 showcase performance variations based on different subsets of the ISIC 2019 dataset. Table 10 uses the Naturalized 2.4k ISIC 2019 dataset for testing, while Table 11 relies on images solely from the Original ISIC 2019 dataset. Overall, both tables exhibit minor discrepancies in precision, recall, and F1-score across various classes. However, Table 11 demonstrates slightly higher accuracy (0.97) compared to Table 10 (0.95), indicating improved performance with the exclusive use of the Original ISIC 2019 dataset for testing. These differences underscore the impact of dataset selection on model evaluation in skin lesion classification tasks.

Table 12 displays the classification report of the fine-tuned DenseNet201 model using the Naturalized 7.2K ISIC 2019 dataset.

**Table 12.** DenseNet201—classification report for the Naturalized 7.2K ISIC 2019.

| Class | Precision | Recall | F1-Score | Support |
|---|---|---|---|---|
| AK | 1.00 | 0.98 | 0.99 | 760 |
| BCC | 1.00 | 1.00 | 1.00 | 760 |
| BK | 1.00 | 1.00 | 1.00 | 760 |
| DER | 1.00 | 1.00 | 1.00 | 760 |
| NEV | 0.98 | 1.00 | 0.99 | 760 |
| MEL | 1.00 | 1.00 | 1.00 | 760 |
| VAS | 1.00 | 1.00 | 1.00 | 760 |
| SCC | 1.00 | 1.00 | 1.00 | 760 |
| Accuracy | | | 1.00 | 5760 |
| Macro Avg. | 1.00 | 1.00 | 1.00 | 5760 |
| Weighted Avg. | 1.00 | 1.00 | 1.00 | 5760 |

## 5. Discussion

The Discussion section encapsulates an extensive analysis of experiments centered around the ambitious task of eight-class skin cancer classification. Utilizing a spectrum of models, including renowned pre-trained ImageNet architectures such as ConvNextBase, ConvNeXtLarge, DenseNets, InceptionResNet V2, EfficientNetB0, VGG-19, VGG16, Xception, alongside Vision Transformer (ViT) models, rigorous assessments were conducted, leveraging the formidable ISIC2019 dataset. To address class imbalance, the innovative "Naturalize" augmentation technique was introduced, resulting in the development of two balanced datasets: Naturalized 2.4K ISIC2019 and Naturalized 7.2K ISIC2019. Quantitative evaluation of the models was executed meticulously through confusion matrices and classification reports, complemented by visual analysis using Score-CAM across four dataset variations: original ISIC2019, Pruned 2.4K ISIC2019, Naturalized 2.4K ISIC2019, and Naturalized 7.2K ISIC2019 datasets.

### 5.1. Naturalized 2.4K ISIC2019 Dataset Results

Initially, transfer learning was employed across all pre-trained models, but a significant improvement was observed upon fine-tuning. Table 6 illustrates the accuracy scores across training, validation, and testing subsets of the Naturalized 2.4K ISIC2019 dataset for fine-tuned models. Notably, the DenseNet201 model exhibited the highest validation and testing accuracies, while the ConvNexTBase model achieved the highest training accuracy among the models. The macro-average precision, recall, and F1-score for the testing subset of the Naturalized 2.4K ISIC2019 dataset, presented in Table 7, reinforced the superiority of the DenseNet-201 model in delivering the most promising results. Given its outstanding performance in the validation and testing subsets, the DenseNet-201 model was selected for subsequent trials.

### 5.2. DenseNet-201 Results

Tables 8–12 portray the classification reports of the fine-tuned DenseNet-201 model using various datasets: original ISIC2019, updated ISIC2019, Naturalized 2.4K ISIC2019, and Naturalized 7.2K ISIC2019, respectively.

Moreover, the success of Naturalize in generating a multitude of high-quality images, mimicking the original dataset, underscores its potential application not only in medical but also in non-medical domains. This triumph showcases Naturalize's adeptness in addressing class imbalance issues, thereby augmenting model performance across diverse classification tasks.

Table 13 offers a holistic view of the DenseNet-201 model's performance across all ISIC 2019 datasets (original, Pruned 2.4K, Naturalized 2.4K and 7.2K), highlighting substantial improvements achieved through dataset balancing. The transition from imbalanced datasets to balanced ones markedly elevated macro-average precision, recall, F1-score, and accuracy. Particularly, the Naturalized 7.2K ISIC2019 dataset displayed exemplary

outcomes, with the DenseNet-201 model achieving perfect scores across all metrics. This underscores the effectiveness of the "Naturalize" augmentation technique in significantly enhancing classification accuracy for identifying skin cancer.

**Table 13.** DenseNet201—classification reports' summaries for all ISIC 2019 datasets (original, Pruned 2.4K, Naturalized 2.4K and 7.2K).

| PBC Datasets | Macro Average | | | |
| --- | --- | --- | --- | --- |
| | Precision | Recall | F1-Score | Accuracy |
| *Imbalanced ISIC 2019 Datasets* | | | | |
| Original | 0.76 | 0.68 | 0.70 | 0.93 |
| Pruned | 0.69 | 0.74 | 0.68 | 0.82 |
| *Naturalized Balanced ISIC 2019 Datasets* | | | | |
| 2.4K (Testing dataset from Naturalized 2.4K ISIC 2019) | 0.96 | 0.95 | 0.95 | 0.96 |
| 2.4K (Testing dataset sourced from Original ISIC 2019) | 0.97 | 0.97 | 0.97 | 0.97 |
| 7.2K | 1.00 | 1.00 | 1.00 | 1.00 |

*5.3. Score-CAM Interpretability*

These findings were further reinforced and expounded upon through the application of Score-CAM, an interpretability technique enabling visualization and comprehension of the model's decision-making process. Figure 10 presents a visual representation derived from Score-CAM, offering an insightful portrayal of the fine-tuned pre-trained DenseNet201 model's performance using the Naturalized ISIC2019 dataset. This visualization not only validates the model's accurate classifications but also transparently delineates the influential regions within the images that contributed to the model's decisions. Score-CAM not only reaffirms the model's exceptional performance but also provides valuable insights into the specific image areas crucial for classification, enriching our understanding of the skin cancer classification process.

*5.4. Comparison with the Previous Works*

Table 14 compares the performance metrics of various previous works alongside our approach in skin cancer classification using the ISIC2019 dataset. Prior research demonstrates a range of accuracies, recall, precision, and F1-scores, showcasing varied results. Our methodology stands out significantly, achieving a perfect score of 100% across all metrics—accuracy, recall, precision, and F1-score. This exceptional outcome signifies a groundbreaking advancement in skin cancer classification, underscoring the effectiveness and reliability of our approach compared to existing methods.

**Table 14.** Comparison with previous works.

| Ref. | Model and Approach | Dataset | Split Ratio | Accuracy | Recall | Precision | F1-Score |
| --- | --- | --- | --- | --- | --- | --- | --- |
| [11] | GoogleNet (Inception V1) and Transfer Learning | ISIC2019 | 80/10/10 | 94.92 | 79.80 | 80.36 | 80.07 |
| [12] | Ensemble CNN-EfficientNet | ISIC2019 | 75/25 | 89.5 | 89.5 | 89.5 | 89.5 |
| [13] | Ensemble Inception-ResNet | ISIC2019 | 60/20/20 | 96.72 | 95.47 | 84.70 | 89.76 |
| [14] | Quantum Inception-ResNet-V1 | ISIC2019 | 80/10/10 | 98.76 | 98.26 | 98.40 | 98.33 |
| [15] | MobileNet and Transfer Learning | ISIC2019 | 80/10/10 | 83 | 83 | 83 | 82 |
| [16] | Wide ShuffleNet and Segmentation | ISIC2019 | 90/10 | 84.80 | 70.71 | 75.15 | 72.61 |
| [17] | Four-layer DCNN | ISIC2019 | 60/10/30 | 84.80 | 83.80 | 80.50 | 81.60 |
| [18] | Residual Deep CNN Model | ISIC2019 | 70/15/15 | 94.65 | 70.78 | 72.56 | 71.33 |
| [19] | Modified EfficientNetV2 | ISIC2019 | 80/20 | 95.49 | 95 | 96 | 95 |
| [20] | DSCC-Net with SMOTE Tomek | ISIC2019 | 80/10/10 | 94.17 | 94.28 | 93.76 | 93.93 |
| [21] | MSCDNet Model | ISIC2019 | 70/20/10 | 98.77 | 98.42 | 98.56 | 98.76 |
| Ours | FT DenseNet201 | Naturalized 7.2K | 80/10/10 | 100 | 100 | 100 | 100 |

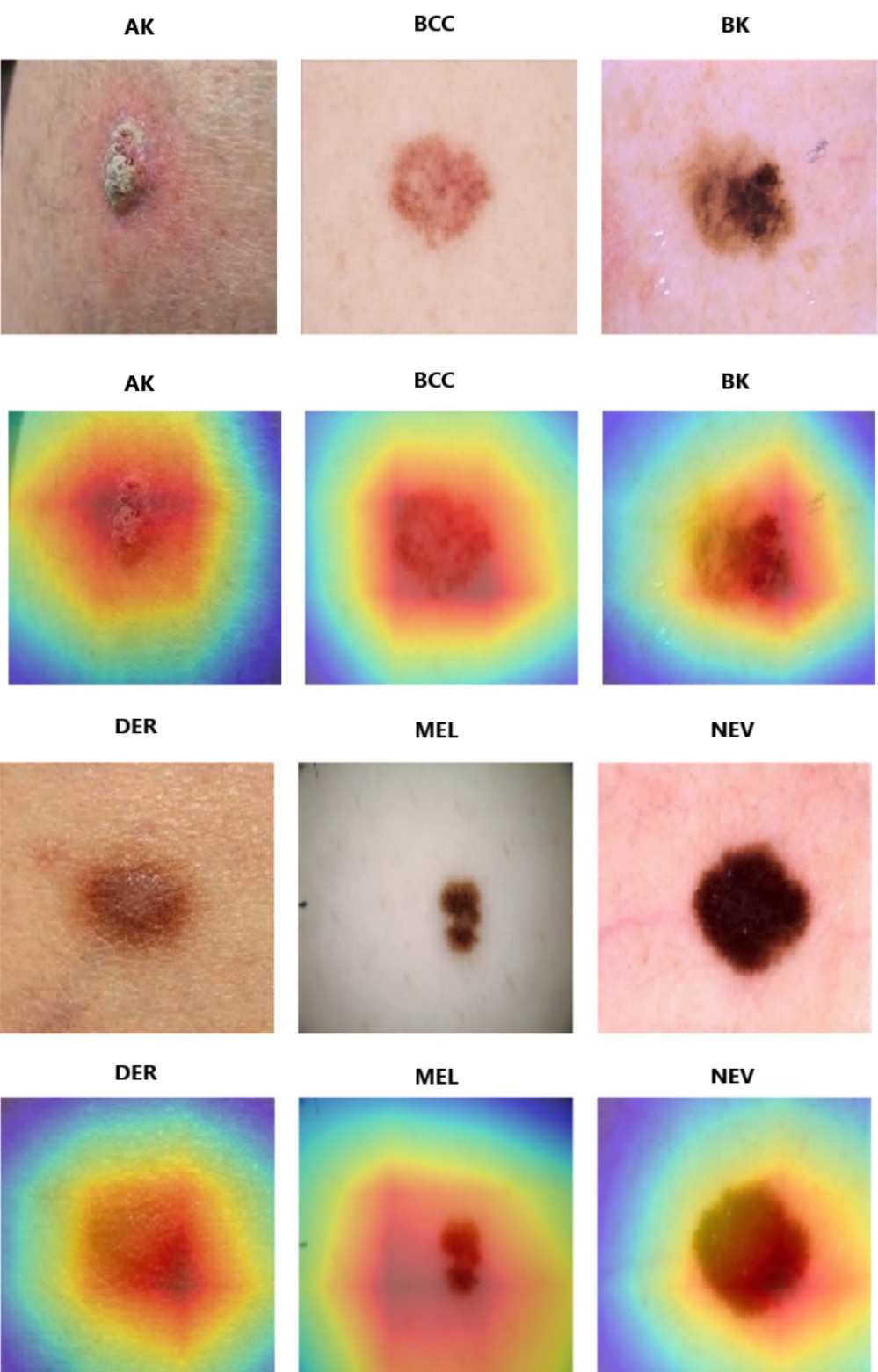

**Figure 10.** Score-CAM for fine-tuned DenseNet-201.

*5.5. Discussion Summary*

In summary, our exploration of eight-class skin cancer classification has yielded compelling results through meticulous analysis of pre-trained models on the challenging ISIC2019 dataset. The introduction of the innovative "Naturalize" augmentation tech-

nique, addressing class imbalance, has proven pivotal in enhancing model performance. The DenseNet-201 model emerged as a standout performer, achieving remarkable accuracy and precision across various datasets.

The fine-tuned DenseNet-201 exhibited superior performance, especially on the Naturalized 7.2K ISIC2019 dataset, attaining perfect scores in all metrics. This dataset, generated by "Naturalize", demonstrated the effectiveness of our augmentation technique in significantly elevating classification accuracy. Interpretability analysis using Score-CAM not only validated the model's decisions but also provided insights into crucial regions influencing classifications.

Comparing our approach with previous works, our methodology stands out with a groundbreaking achievement of 100% accuracy, recall, precision, and F1-score. This underscores the robustness and reliability of our model, setting a new benchmark for performance in skin cancer classification.

In conclusion, the success of "Naturalize" in generating high-quality images has far-reaching implications, not only in the medical domain but also in broader applications. Our approach not only addresses the challenges of skin cancer classification but also sets a new benchmark for performance, emphasizing the transformative impact of innovative augmentation techniques in enhancing the capabilities of deep learning models.

## 6. Conclusions

This study delved into the challenges of skin cancer diagnosis, traditionally hindered by subjectivity and resource constraints. Leveraging Artificial Intelligence (AI) for eight-class skin cancer classification, our research utilized advanced deep learning models on the ISIC2019 dataset. Noteworthy contributions include the introduction of the "Naturalize" augmentation technique, addressing class imbalances and leading to the creation of the high-impact Naturalized 7.2K ISIC2019 dataset. The pivotal role of AI in mitigating misdiagnosis risks and enhancing dermatological diagnostics cannot be overstated. Our meticulous evaluations, culminating in 100% average accuracy, precision, recall, and F1-score within the Naturalized 7.2K ISIC2019 dataset, underscore the transformative potential of AI-driven methodologies. This research signifies a paradigm shift in dermatological diagnosis, advocating for the integration of AI-driven solutions into clinical practice. The perfect performance within the Naturalized 7.2K ISIC2019 dataset signals a new era in skin cancer care, emphasizing the urgency of adopting AI-driven methodologies for improved diagnostic precision and patient outcomes.

**Author Contributions:** Conceptualization, M.A.A. and F.D.; methodology, M.A.A., F.D. and H.A.; software, M.A.A.; validation, M.A.A.; formal analysis, M.A.A., F.D. and I.A.-C.; investigation, M.A.A.; resources, M.A.A. and F.D.; data curation, M.A.A.; writing—original draft preparation, M.A.A., F.D., H.A. and M.K.; writing—review and editing, M.A.A., F.D. and I.A.-C.; supervision, F.D. and I.A.-C.; project administration, F.D. and I.A.-C.; funding acquisition, F.D. and I.A.-C. All authors have read and agreed to the published version of the manuscript.

**Funding:** This work is partially supported by grant GIU23/022 unded by the University of the Basque Country (UPV/EHU), and grant PID2021-126701OB-I00, funded by the Ministerio de Ciencia, Innovación y Universidades, AEI, MCIN/AEI/10.13039/501100011033, and by "ERDF A way of making Europe" (to I.A-C.)

**Institutional Review Board Statement:** Not applicable.

**Informed Consent Statement:** Not applicable.

**Data Availability Statement:** The data usedin this paper are publicly available.

**Conflicts of Interest:** The authors declare that the research was conducted in the absence of any commercial or financial relationships that could be construed as a potential conflict of interest.

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
