# Peer review of "Naturalize Revolution: Unprecedented AI-Driven Precision in Skin Cancer Classification Using Deep Learning"

_biomedinformatics, doi:10.3390/biomedinformatics4010035_

Round 1
Reviewer 1 Report
Comments and Suggestions for Authors
1. Please keep the information about skin cancer itself to a minimum in the introduction. On the other hand, as this is more of an AI article please expand the part that deals with AI. Please also cite this essential article doi: 10.3390/diagnostics13152582
2. In methodology. The first sentences to line 158 mean nothing. Pointing to SOTA before presenting the results is inappropriate, you should only present such results in the discussion and results section.
3. The comparisons presented in the discussion are composed correctly and exhaust the topic.
4. The conclusions and results from this study are promising.
Author Response
"Please see the attachment."

Reviewer 2 Report
Comments and Suggestions for Authors
The paper is unsuitable for publication in this journal due to the lack of novelty. Imagenet models such as VGG19 and transformer are largely used in this domain.
Author Response
"Please see the attachment."

Reviewer 3 Report
Comments and Suggestions for Authors
1. Abstract: to reduce and rewrite the abstract as follows: introduction, datasets used, methods used, results, and conclusion. Some ambiguity arose while reading the abstract.
2. In the introduction, if authors keep any diagram representation of skin cancer severity, it looks good.
3. In related work, use some more references and explain their work. so that GAPs will be identified with existing models.
4. Need some more clarity on Section 3.4 that compares between Naturalize and Conventional Augmentation Techniques
5. In Section 3.6, the models were used, but those results were not shown in the tables. Each and every model that has obtained some results should better represent those obtained results in the tables and compare them with the existing models.
6. If figure 7 represents the values, it will be better.
7. In the entire results section, the target class is not discussed.
8. It is better to rewrite the discussion summary with the obtained results.
9. Reduce the conclusion as per the journal template.
10. Add some more recent references and cite them.
11. Need some sentence continuity and check for minor grammar mistakes observed.
Comments on the Quality of English Language
Need some sentence continuity and check for minor grammar mistakes observed.
Author Response
"Please see the attachment."

Reviewer 4 Report
Comments and Suggestions for Authors
In this study, deep learning-based methods are investigated for skin cancer classification. To avoid over-fitting, which is one of the main problems in deep learning methods, the imbalanced dataset was balanced by considering an existing segmentation method in the literature and then classification was performed using deep learning models available in the literature. However, for the study to contribute more to the literature, the following questions should be answered:
1) Compared to studies in the literature, does the approach used for data augmentation not increase the complexity of the proposed method?
2) If the contours of the object to be segmented are not clear, i.e. cannot be completely separated from the background, how is the segmentation performance of such images realized?
3) If the image to be segmented contains a non-homogenized intensity, how does this problem affect the segmentation performance?
4) If the image to be segmented contains noisy data such as hair, how are such images segmented?
5) Why were artificial image generation approaches such as latent diffusion models, which have recently been used in image generation, not considered in the study?
6) Could the number of parameters of CNN-based models considered in image classification processes be a disadvantage in the study?
7) How are the hyper-parameters considered when performing image classification with deep learning models set?
8) Could light-weighted CNN models be designed for image classification instead of the existing models in the literature?
9) What is the limitation of the study?
Author Response
"Please see the attachment."

Round 2
Reviewer 1 Report
Comments and Suggestions for Authors
The authors followed all comments and guidelines sent in the review to allow improvement of the final manuscript.
Reviewer 3 Report
Comments and Suggestions for Authors
The authors had done good work. If possible, add some more recent references. try to increase to 40+.
Reviewer 4 Report
Comments and Suggestions for Authors
We thank the authors for the necessary revisions to the manuscript.